# Thermodynamic Bayesian Inference

## Abstract

A fully Bayesian treatment of complicated predictive models (such as deep neural networks) would enable rigorous uncertainty quantification and the automation of higher-level tasks including model selection. However, the intractability of sampling Bayesian posteriors over many parameters inhibits the use of Bayesian methods where they are most needed. Thermodynamic computing has emerged as a paradigm for accelerating operations used in machine learning, such as matrix inversion, and is based on the mapping of Langevin equations to the dynamics of noisy physical systems. Hence, it is natural to consider the implementation of Langevin sampling algorithms on thermodynamic devices. In this work we propose electronic analog devices that sample from Bayesian posteriors by realizing Langevin dynamics physically. Circuit designs are given for sampling the posterior of a Gaussian-Gaussian model and for Bayesian logistic regression, and are validated by simulations. It is shown, under reasonable assumptions, that the time-complexity of sampling the Gaussian-Gaussian posterior is sublinear in dimension. These results highlight the potential to accelerate Bayesian inference with thermodynamic computing.

## 1 Introduction

Bayesian statistics has proved an effective framework for making predictions under uncertainty [1, 2, 3, 4, 5, 6], and it is central to proposals for automating machine learning [7]. Bayesian methods enable uncertainty quantification by incorporating prior knowledge and modeling a distribution over the parameters of interest. Popular machine learning methods that employ this approach include Bayesian linear and non-linear regression [8], Kalman filters [9], Thompson sampling [2], continual learning [10, 11], and Bayesian neural networks [3, 12].

Unfortunately, computing the posterior distribution in these settings is often intractable [13]. Methods such as the Laplace approximation [14] and variational inference [15] may be used to approximate the posterior in these cases, however their accuracy struggles for complicated posteriors, such as those of a Bayesian neural network [13]. Regardless, sampling accurately from such posteriors requires enormous computing resources [13].

Computational bottlenecks in Bayesian inference motivate the need for novel hardware accelerators. Physics-based sampling hardware has been proposed for this purpose, including Ising machines [16, 17, 18, 19, 20], probabilistic bit computers [21, 22, 23], and thermodynamic computers [24, 25, 26, 27, 28, 29]. Continuous-variable hardware is particularly suited to Bayesian inference since continuous distributions are typically used in probabilistic machine learning [30]. However, a rigorous treatment of how such hardware can perform Bayesian inference with scalable circuits has not yet been given.

Submitted to the Second Workshop on Machine Learning with New Compute Paradigms at NeurIPS (MLNCP 2024). Do not distribute.

The most computationally tractable algorithms for exact Bayesian inference are Monte Carlo sampling algorithms. The Langevin sampling algorithm [31, 32] is an elegant example inspired by statistical physics, based on the dynamics of a damped system in contact with a heat bath. What we propose in this work is to build a physical realization of the system that is simulated by the Langevin algorithm. The system must be designed to have a potential energy such that the Gibbs distribution $p(x) \propto e^{-\beta U(x)}$ is the desired posterior distribution which is reached at thermodynamic equilibrium. We present circuit schematics for electronic implementations of such devices for Bayesian inference for two special cases. The first is a Gaussian-Gaussian model (where the prior and the likelihood are both multivariate normal, as found in linear regression and Kalman filtering), and the second is logistic regression (where the prior is Gaussian and the likelihood is Bernoulli parameterized by a logistic function). In each case, the parameters of the prior and likelihood are encoded in the values of components of the circuit, and then voltages or currents are measured to sample the random variable.

While thermodynamic algorithms have been proposed for linear algebra [27] and neural network training [33], our work can be viewed as the first thermodynamic algorithm for sampling from Bayesian posteriors. Moreover, our work provides the first concrete proposal for non-Gaussian sampling with thermodynamic hardware. Overall, our work opens up a new field of rigorous Bayesian inference with thermodynamic computers and lays the groundwork for scalable CMOS-based chips for probabilistic machine learning.

We show that in theory the device proposed for sampling the Gaussian-Gaussian model posterior can obtain $N$ samples in $d$ dimensions in time scaling with $O(N \ln d)$. This is a significant speedup over typical methods used digitally for the same problem, which involve matrix inversions taking time scaling with $O(d^\omega)$ where $2 < \omega < 3$. This speedup is larger than the polynomial speedups found in previous work on thermodynamic algorithms for linear algebra primitives [27] (where speedups were found to scale linearly with dimension).

## 2   Results

Suppose that we have samples of a random vector $y$, and would like to estimate a random vector $\theta$ on which $y$ depends somehow. The Bayesian approach is to assume a prior distribution on $\theta$ given by a density function $p_\theta(\theta)$, and a likelihood function $p_{y|\theta}(y|\theta)$. The posterior distribution for $\theta$ is then given by Bayes's theorem $p_{\theta|y}(\theta|y) = p_{y|\theta}(y|\theta)p_\theta(\theta)/p_y(y)$. To sample from the posterior using the Langevin algorithm, one first computes the score

$$\nabla_\theta \ln p_{\theta|y}(\theta|y) = \nabla_\theta \ln p_{y|\theta}(y|\theta) + \nabla_\theta \ln p_\theta(\theta). \tag{1}$$

Then the score is used as the drift term in the following stochastic differential equation (SDE)

$$d\theta = \nabla_\theta \ln p_{\theta|y}(\theta|y)\, dt + \mathcal{N}[0, 2\, dt]. \tag{2}$$

After this SDE is evolved for a sufficient time $T$, the value of $\theta$ will be a sample from $p_{\theta|y}$. This algorithm is equivalent to the equilibration of an overdamped system, as we will now describe. First let $r$ be a vector of the same dimension as $\theta$ describing the state of a physical system, and satisfying $r = \theta\tilde{r}$ for some constant $\tilde{r}$ (this factor is necessary because $\theta$ is unitless while the physical quantity $r$ has units). Now we define the potential energy function $\beta U(r) = -\ln p_{\theta|y}(r/\tilde{r} \mid y)$. The dynamics of an overdamped system with potential energy $U$ in contact with a heat bath at inverse temperature $\beta$ can be modeled by the overdamped Langevin equation

$$dr = -\gamma^{-1}\nabla_r U(r)\, dt + \mathcal{N}[0, 2\gamma^{-1}\beta^{-1}\, dt], \tag{3}$$

where $\gamma$ is a damping constant. Note that his implies that $\gamma$ has dimensions of energy $\cdot$ time$/[r]^2$. If we introduce a constant $\tau = \gamma\beta\tilde{r}^2$, Eq. (3) can be written

$$d\theta = \nabla_\theta \ln p_{\theta|y}(\theta|y)\tau^{-1}\, dt + \mathcal{N}[0, 2\,\tau^{-1}dt], \tag{4}$$

which has the same form as Eq. (2), except with the time constant $\tau$. It is clear that if Eq. (2) must be run for a dimensionless duration $T$ to achieve convergence, then the physical system must be allowed to evolve for a physical time duration $\tau T$ to achieve the same result. While we have addressed the case of conditioning on a single sample $y$ above, the generalization of these ideas to the case of conditioning on multiple I.I.D. samples is given in Appendix D. In what follows we will present designs for circuits whose potential energy results in an overdamped Langevin equation that yields samples from Bayesian posteriors.

## 2.1 Gaussian-Gaussian model

A particularly simple special case of Bayesian inference is a when both the prior and the likelihood are multivariate normal, and we address this simple model first in order to illustrate our approach more clearly. Specifically, let $\theta \in \mathbb{R}^d$ have prior distribution $p_\theta(\theta) = \mathcal{N}[\mu, \Sigma]$, and let the likelihood be $p_{y|\theta}(y|\theta) = \mathcal{N}[\theta, \Sigma_{y|\theta}]$, where $y \in \mathbb{R}^d$ is an observed sample. In this case the posterior $p_{\theta|y}$ is also multivariate normal, with parameters [12]

$$\mu_{\theta|y} = \mu + \Sigma \left( \Sigma + \Sigma_{y|\theta} \right)^{-1} (y - \mu), \tag{5}$$

$$\Sigma_{\theta|y} = \Sigma - \Sigma(\Sigma + \Sigma_{\theta|y})^{-1}\Sigma. \tag{6}$$

For this model, the posterior is tractable and can be computed on digital computers relatively efficiently, however for very large dimensions the necessary matrix inversion and matrix-matrix multiplications can still create a costly computational bottleneck. As we will see, the thermodynamic approach provides a means to avoid the costly inversion and matrix products in the computation, and therefore to accelerate Bayesian inference for this model.

We begin by deriving the Langevin equation for sampling this posterior. For this prior and likelihood, the score of the posterior Eq. (1) is

$$\nabla_\theta \ln p_{\theta|y}(\theta|y) = -\Sigma^{-1}(\theta - \mu) - \Sigma_{y|\theta}^{-1}(\theta - y), \tag{7}$$

and so Eq. (4) becomes

$$d\theta = -\Sigma^{-1}(\theta - \mu)\tau^{-1}dt - \Sigma_{y|\theta}^{-1}(\theta - y)\tau^{-1}dt + \mathcal{N}[0, 2\mathbb{I}\tau^{-1}dt]. \tag{8}$$

In fact, this SDE can be implemented by a circuit consisting of two resistor networks coupled by inductors, shown in Fig. 1 for the two-dimensional case.

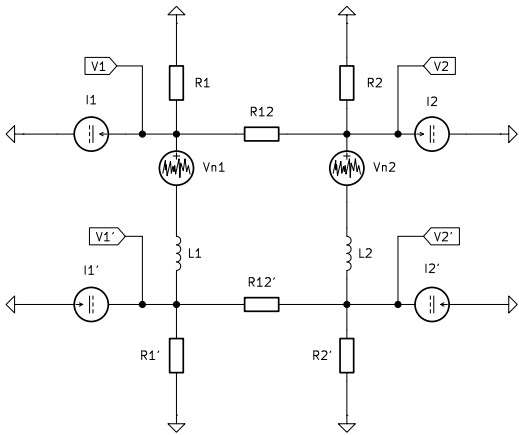

Figure 1: Circuit schematic for the Gaussian-Gaussian model posterior sampling device.

The full analysis of the circuit in Fig. 1 is given in Appendix A, but a few remarks are made here to explain its operation. First, we define the conductance matrices $\mathcal{G}$ as

$$\mathcal{G} = \begin{pmatrix} R_{11}^{-1} + R_{12}^{-1} & -R_{12}^{-1} \\ -R_{12}^{-1} & R_{22} + R_{12}^{-1} \end{pmatrix}, \tag{9}$$

and $\mathcal{G}'$ is defined in the same way for the primed resistors $R_1'$, $R_2'$, and $R_{12}'$. By applying Kirchoff's current law (KCL), the voltages across the resistors can be eliminated. Then the equation $V = L\dot{I}$ is used to derive the following stochastic differential equation for the currents through the inductors

$$dI_L = -L^{-1}\mathcal{G}^{-1}(I_L - I)\,dt - L^{-1}\mathcal{G}'^{-1}(I_L - I')\,dt + L^{-1}\sqrt{S}\mathcal{N}[0, \mathbb{I}\,dt], \tag{10}$$

where $I_L = (I_{L1} \ I_{L2})^\mathsf{T}$ and $S$ is the power spectral density of each noise source. This equation has the same form as Eq. (8), so it is only necessary to determine an appropriate mapping of distributional parameters to physical properties of the circuit's components (see Appendix A). By

including more inductors and coupling resistors (as well as current and voltage sources), the design can be generalized to arbitrary dimension.

To verify that the proposed circuit does indeed evolve according to the correct SDE, we ran SPICE circuit simulations. Figure 2 shows the results of such a simulation where a 2-dimensional Gaussian prior and a 2-dimensional Gaussian likelihood are encoded into the conductances while the current in each inductor is measured to determine the resulting posterior.

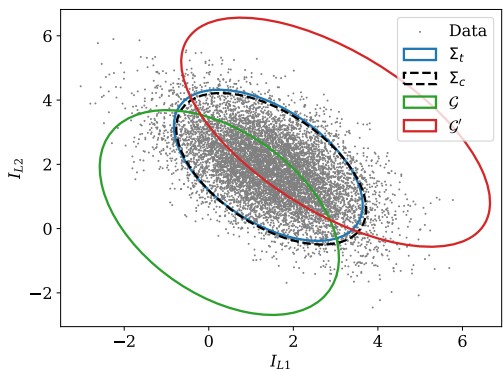

Figure 2: SPICE simulations of proposed Gaussian-Gaussian circuit in Fig. 1. The grey points represent the simulated circuit's induct currents. The dashed black and solid blue ellipses represent the empirical sample covariance and the target posterior covariance from a Gaussian Bayesian update, respectively. The red and green ellipses represent the prior and likelihood.

As shown in Appendix C, the asymptotic runtime complexity for this algorithm is

$$t = O(N\kappa\tau \ln(\kappa^{3/2}d^{1/2}W_0^{-1})), \tag{11}$$

where $\kappa$ is the condition number of the posterior covariance, $\tau = L/\tilde{R}$, $d$ is the dimension, and $W_0$ is the Wasserstein distance between the true posterior and the distribution sampled by the device. The assumptions used to derive this result can also be found in Appendix C. Remarkably, the required time is sublinear in dimension, a large improvement over digital algorithms where complexity of constructing and sampling from the Gaussian-Gaussian posterior (67 - 68) is $O(d^\omega)$ where $\omega$ is the matrix multiplication constant (or more practically $O(d^3)$ via common implementations of Cholesky factorization). In Figure 3(a), we report the convergence of simulated thermodynamic samples for the Gaussian-Gaussian model with zero prior mean and covariances $\Sigma$, $\Sigma_{y|\theta}$ randomly sampled from a Wishart distribution with $2d$ degrees of freedom. We see fast convergence in Wasserstein distance to the true posterior, supporting our theoretical claims.

## 2.2 Bayesian linear regression and Kalman filtering

A generalization of the Gaussian-Gaussian model is that of Bayesian linear regression [8] (or equivalently a Kalman filter update step [9, 12]). In full generality we have

$$p_\theta(\theta) = \mathcal{N}[\mu, \Sigma], \tag{12}$$
$$p_{y|\theta}(y \mid \theta) = \mathcal{N}[H\theta, \Sigma_{y|\theta}], \tag{13}$$

Then the overdamped Langevin SDE becomes

$$d\theta = -\Sigma^{-1}(\theta - \mu)\tau^{-1}\,dt - H^\intercal\Sigma_{y|\theta}^{-1}(y - H\theta)\tau^{-1}\,dt + \mathcal{N}[0, 2\mathbb{I}\tau^{-1}dt],$$
$$= -(A\theta - b)\tau^{-1} + \mathcal{N}[0, 2\mathbb{I}\tau^{-1}dt], \text{ for } A = \Sigma^{-1} + H^\intercal\Sigma_{y|\theta}^{-1}H \text{ and } b = \mu + H^\intercal\Sigma_{y|\theta}^{-1}y. \tag{14}$$

The form of the SDE (Ornstein-Unhlenbeck process) in 14 is exactly that of the thermodynamic device in [27] which if given input $A$ and $b$ above will produce samples from the Gaussian Bayesian posterior $p_{\theta|y}(\theta \mid y)$. Compared to the simpler Gaussian-Gaussian model above, a disadvantage of this approach is thathe covariances $\Sigma$ and $\Sigma_{y|\theta}$ have to be inverted prior to input as $A$. However, for linear regression, these matrices are often assumed to be diagonal and otherwise they can be

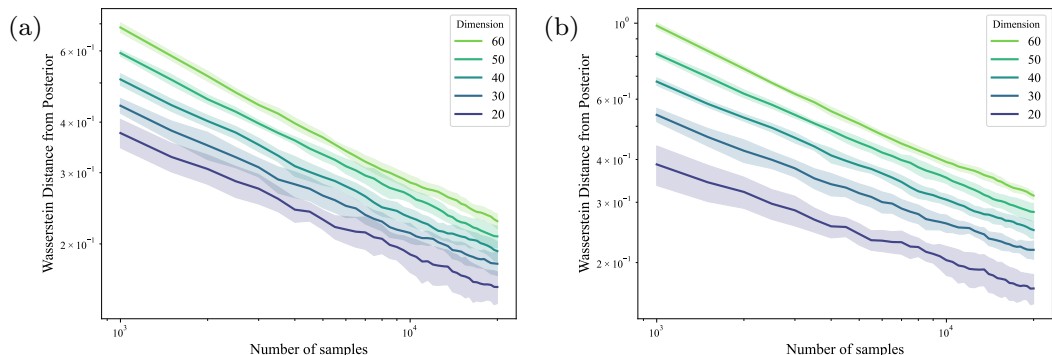

Figure 3: Convergence in Wasserstein distance between simulated thermodynamic samples and the true Gaussian posterior as a function of the number of samples (sampling time). All results are simulated exactly with `thermox` [35] and averaged over 10 random seeds with one standard deviation shown. Panel (a): Gaussian-Gaussian model with zero prior mean and covariances sampled from a Wishart distribution. Panel (b): Bayesian linear regression on the diabetes dataset [36] with dimension (number of features) varied by including higher-order cross terms of the 10 input data features.

efficiently inverted using the thermodynamic procedures in [27] as preprocessing. Additionally, the formulation of $A$ requires matrix-matrix multiplications which can be costly (even in the case of diagonal covariances). Although, this can be accelerated with parallelization.

On the other hand, the generality of (12-13) makes the approach highly practical. Encompassing Bayesian linear regression [34] and the update step of the Kalman filter [9]. Moreover in the setting of Kalman filtering, the matrices $\Sigma$ and $\Sigma_{y|\theta}$ are typically shared across time points and thus only need to be inverted once in comparison to the Bayesian posterior update which is applied at every time step (and typically represents the computation bottleneck due to the required matrix inversion).

In Figure 3(b), we simulate the evaluation of the thermodynamic linear algebra device [27] for a Bayesian linear regression task. We use the diabetes dataset [36] which has $N = 442$ continuous response variables $y$ and 10 input features. We vary the number of features and therefore posterior dimension for the linear regression by extending to include the first $d$ cross terms in the Taylor expansion over the input features. These input features are loaded as rows in the matrix $H \in \mathbb{R}^{N \times d}$. Both covariances are set to diagonal, $\Sigma = \mathbb{I}$ and $\Sigma_{y|\theta} = 0.1\mathbb{I}$. We observe that the Wasserstein distance converges quickly as more samples are collected and scales reasonably with dimension, indicating a sublinear scaling similar to the Gaussian-Gaussian model.

## 2.3 Bayesian logistic regression

Logistic regression is a method for classification tasks (both binary and multiclass) that models the dependence of class probabilities on independent variables using a logistic function. In the Bayesian setting, a prior can be assumed on the parameters of a logistic regression model, for example it is common to assume a Gaussian prior. However, after conditioning on observed data a posterior distribution is produced that has no analytical closed form, making Bayesian logistic regression far less efficient than obtaining a point estimate of the parameters. In this section we present a thermodynamic hardware architecture capable of sampling the posterior for binary logistic regression, and show some preliminary evidence that this architecture can do so more efficiently than existing methods.

Given a parameter vector $\theta \in \mathbb{R}^d$ and an independent variable vector $x \in \mathbb{R}^d$, binary logistic regression outputs a class probability $p_{y|\theta,x}(y|\theta,x)$, where $y \in \{-1, 1\}$ (often $y \in \{0, 1\}$ is written instead but we choose this notation to simplify the presentation). The likelihood is $p_{y|\theta,x}(y|\theta,x) = L(y\theta^\mathsf{T} x)$ where $L(z) = 1/(1 + e^{-z})$ is the standard logistic function [37]. Note that we will first consider the case of conditioning on a single sample, and in this case the likelihood will be denoted $p_{y|\theta}(y|\theta)$ as $x$ is constant. Additionally, a multivariate normal prior is assumed for the parameters $\theta \sim \mathcal{N}[\mu, \Sigma]$. The Langevin equation for sampling the posterior is therefore:

$$d\theta = -\Sigma^{-1}(\theta - \mu)\tau^{-1}dt + L(-y\theta^\mathsf{T} x)yx\tau^{-1}dt + \mathcal{N}[0, 2\mathbb{I}\tau^{-1}dt]. \tag{15}$$

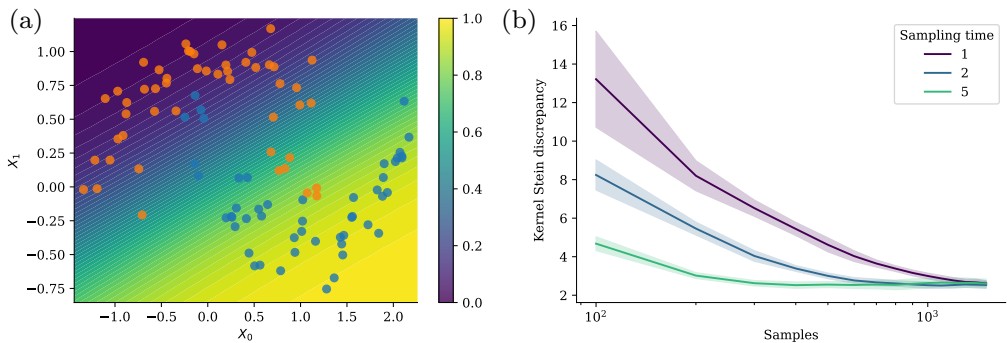

Figure 4: Panel (a): Probability surface to belong to class 1 (blue points). The dataset is also shown, where class 0 (blue points) and class 2 (orange points) are arranged in two intersecting moons. Panel (b): Kernel Stein discrepancy (KSD) of the collected samples with an ideal thermodynamic sampler, for varying sampling times. The sampling time is given in units of $10^{-3}\tau$. The KSD is averaged over five sets of random samples and $\tau = 1$.

A circuit implementing Eq. (15) is shown in Fig. 7, and the detailed analysis of this circuit is given in Appendix B. Equation (15) is valid for a single data sample, however, as mentioned, in practice we generally take gradients over a larger number of examples such that the gradients are less noisy. This can be done by enlarging the hardware, resulting in the second term of Eq. (15) being replaced by a sum $\sum_{i=1}^{N} L(-y_i\theta^\intercal x_i)y_i x_i dt$, with $N$ the number of data points. One may also consider minibatches, and the sum is only over a batch of size $b$. This is achievable by summing currents, which is detailed in the circuit implementation in Appendix B. At a high-level, implementing this protocol in hardware is very simple in the case of a full batch, since the data only needs to be sent once onto the hardware. The following steps are taken to collect the samples: (1) Map the data labels to $\{+1, -1\}$. (2) Map the data $(X, Y)$ onto the hardware (full batch setting). (3) Initialize the state of the system, set the mean and the covariance matrix of the prior. (4) At every interval $t_s$ (the sampling time), measure the state of the system $\theta(t)$ to collect samples.

In Fig. 4, we present results for a Bayesian logistic regression on a two-moons dataset, made of points separated in two classes that are arranged in intersecting moons in the 2D planes, as shown in Fig. 4(a). These results are obtained by running the SDE of Eq. (15), hence corresponds to an ideal simulation of the thermodynamic hardware. In this scenario, there are 3 parameters to sample, and $N = 100$ points are considered. In Fig. 4(a), we see that even for such a simple model, only a few points are missclassified. As mentioned, previously, this setting also gives access to better methods to estimate uncertainty in predictions. In Fig. 4(b), the Kernel Stein discrepancy (KSD) [38] is shown as a function of the number of collected samples for varying sampling rates. These results indicate that the number of samples to reach a low KSD (close to convergence) can be reduced by increasing the sampling time, indicating correlated samples, as is often the case.

## 3   Conclusion

In this work, we proposed the first thermodynamic algorithms for sampling from Bayesian posteriors. We provided explicit constructions of CMOS-compatible analog circuits to implement these algorithms with scalable silicon chips. Our circuit for performing logistic regression represents the first concrete proposal for non-Gaussian sampling with a thermodynamic computer. In the case of Gaussian Bayesian inference (Gaussian prior, Gaussian likelihood), our analysis showed a sublinear complexity in $d$, leading to a speedup over standard digital methods that is greater than linear. This is an even larger speedup than those previously observed for thermodynamic linear algebra [27], suggesting that Bayesian inference is an ideal application for thermodynamic computers. Our work lays the foundation for accelerating Bayesian inference, a key component of probabilistic machine learning, with physics-based hardware.

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

# A   Analysis of Gaussian Bayesian Inference Circuit

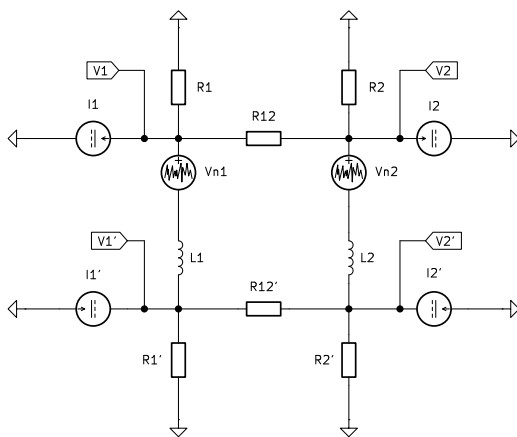

Figure 5: Circuit schematic for the Gaussian-Gaussian model posterior sampling device.

In Figure 5, positive current goes up through the two inductors, left to right through $R_{12}$ and $R'_{12}$, and towards ground in the other resistors. The two inductors have the same inductance $L$. KCL gives

$$I_{L1} - I_1 = I_{R1} + I_{12} \tag{16}$$

$$I_{L2} - I_2 = I_{R2} - I_{12} \tag{17}$$

$$-I_{L1} + I'_1 = I'_{R1} + I'_{12} \tag{18}$$

$$-I_{L2} + I'_2 = I'_{R2} - I'_{12}. \tag{19}$$

Using Ohm's law,

$$I_{L1} - I_1 = R_1^{-1}V_1 + R_{12}^{-1}(V_1 - V_2) = (R_1^{-1} + R_{12}^{-1})V_1 - R_{12}^{-1}V_2 \tag{20}$$

$$I_{L2} - I_2 = R_2^{-1}V_2 - R_{12}^{-1}(V_1 - V_2) = (R_2^{-1} + R_{12}^{-1})V_2 - R_{12}^{-1}V_1. \tag{21}$$

These can be written as a single vector equation as follows

$$I_L - I = \mathcal{G}V, \tag{22}$$

where $I_L = (I_{L1}\ I_{L2})^\intercal$, $I = (I_1\ I_2)^\intercal$, and

$$\mathcal{G} = \begin{pmatrix} R_1^{-1} + R_{12}^{-1} & -R_{12}^{-1} \\ -R_{12}^{-1} & R_2^{-1} + R_{12}^{-1} \end{pmatrix}. \tag{23}$$

Similarly, for the lower subcircuit we have

$$-I_L + I' = \mathcal{G}'V'. \tag{24}$$

The inductors obey the equations

$$L_1\dot{I}_{L1} = V'_1 - (V_1 - V_{n1}) \tag{25}$$

$$L_2\dot{I}_{L2} = V'_2 - (V_2 - V_{n2}), \tag{26}$$

or in vector notation

$$L\dot{I}_L = V' - V + V_n. \tag{27}$$

Substituting in the expressions for $V$ and $V'$ derived before, we have

$$L\dot{I}_L = \mathcal{G}'^{-1}(I' - I_L) - \mathcal{G}^{-1}(I_L - I) + V_n, \tag{28}$$

or

$$\dot{I}_L = -L^{-1}\mathcal{G}^{-1}(I_L - I) - L^{-1}\mathcal{G}'^{-1}(I_L - I') + L^{-1}V_n. \tag{29}$$

$$dI_L = -L^{-1}\mathcal{G}^{-1}(I_L - I)\,dt - L^{-1}\mathcal{G}'^{-1}(I_L - I')\,dt + L^{-1}\sqrt{S}\mathcal{N}[0, \mathbb{I}\,dt]. \tag{30}$$

We now proceed to non-dimensionalize the above equation. $\mathcal{G} = \tilde{R}^{-1} A$.

$$\tilde{I} d\theta = -\tilde{I}\tilde{R}L^{-1}A^{-1}(\theta - \mu)\, dt - \tilde{I}\tilde{R}L^{-1}A'^{-1}(\theta - \mu')\, dt + L^{-1}\sqrt{S}\mathcal{N}[0, \mathbb{I}\, dt]. \quad (31)$$

Define $\tau = L/\tilde{R}$, giving

$$d\theta = -A^{-1}(\theta - \mu)\tau^{-1}\, dt - A'^{-1}(\theta - \mu')\tau^{-1}\, dt + \tilde{I}^{-1}L^{-1}\sqrt{S}\mathcal{N}[0, \mathbb{I}\, dt]. \quad (32)$$

If we set $S = 2\tilde{I}^2 L\tilde{R}$, then we have

$$d\theta = -A^{-1}(\theta - \mu)\tau^{-1}\, dt - A'^{-1}(\theta - \mu')\tau^{-1}\, dt + \mathcal{N}[0, 2\mathbb{I}\tau^{-1}\, dt]. \quad (33)$$

# B    Analysis of Bayesian Logistic Regression Circuit

We now analyze the circuit in Figure 7. The boxes labeled Diff. Pair represent differential pairs of NPN bipolar junction transistors (BJTs), as shown in Fig. 6. To achieve a working implementation, additional circuitry is needed to support the differential pair and assure that it is appropriately biased, including a power source and possibly current mirrors.

The following conventions for current flow will be used

- $I_C$ is the current *into* the collector of a transistor. $I_B$ is the current *into* the base of a transistor. $I_E$ is the current *out of* the emitter of a transistor.
- The output current $I_o$ of a differential pair is the current that flows *into* the collector of the BJT labeled $Q_a$.
- Positive current flows in the direction of the arrow through all current sources.
- Positive current flows downwards through $C_1$ and $C_2$ and from left to right through $R_{12}$.
- Through resistors $R_{A11}$, $R_{B11}$, etc. positive current always flows towards the base of the transistor.

## B.1    Analysis of the BJT differential pair

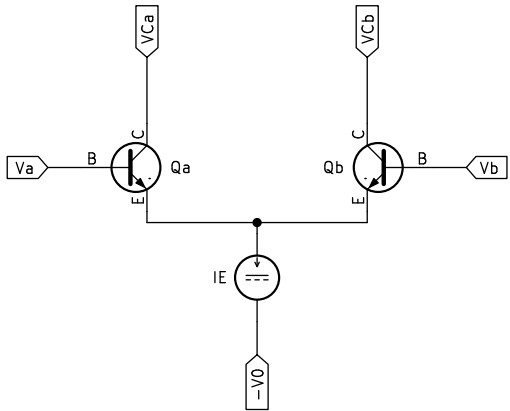

Figure 6: Circuit schematic for the BJT differential pair.

We first consider the behavior of differential pair subcircuit, which can be explained using the Ebers-Moll model. The Ebers-Moll model describes the BJT in active mode, meaning when $V_E < V_B < V_C$, and the circuit must be appropriately biased at all times to ensure the device is always in active mode. According to this model, in active mode the following relations are satisfied

$$I_C = I_S \left( e^{(V_B - V_E)/V_T} - 1 \right), \quad (34)$$

$$I_C = \alpha I_E, \quad (35)$$

where $I_S$ is the saturation current, $V_T$ the thermal voltage, and $\alpha$ is the common-base current gain. $I_S$ is typically on the order of $10^{-15}$ to $10^{-12}$ Amps, and at room temperature $V_T = 25.3\text{mV}$. The parameter $\alpha$ is between 0.98 and 1. It follows from Kirchoff's current law (KCL) that $I_B = (1-\alpha)I_E$. For these typical values of the parameters appearing in Eq. (34) the subtraction of unity in parentheses can safely be ignored, which we will do in what follows. In order for the Ebers-Moll model to be valid, the voltage $V_0$ should be determined such that $V_C > V_B > V_E$ for all transistors at all times, but the value of $V_0$ is otherwise unimportant.

To analyze the differential pair of transistors $Q_a$ and $Q_b$, observe that (by KCL)

$$I_{Ea} + I_{Eb} = I_E. \tag{36}$$

We must distinguish between the two base voltages $V_a$ and $V_b$, but the two emitter voltages are the same, so we write $V_E = V_{Ea} = V_{Eb}$. Using Eqs. (34) and (35) then,

$$I_E = \frac{I_S}{\alpha}e^{-V_E/V_T}\left(e^{V_a/V_T} + e^{V_b/V_T}\right), \tag{37}$$

where we have dropped the $-1$ as explained earlier. Now the emitter current $I_{Ea}$ can be written as

$$I_{Ea} = \frac{I_S}{\alpha}e^{(V_a-V_E)/V_T} \tag{38}$$

$$= \frac{I_E e^{V_a/V_T}}{e^{V_a/V_T} + e^{V_{Bb}/V_T}} \tag{39}$$

$$= \frac{I_E}{1 + e^{-(V_a-V_b)/V_T}}, \tag{40}$$

and similarly

$$I_{Eb} = \frac{I_E}{1 + e^{(V_a-V_b)/V_T}}. \tag{41}$$

Equation (35) is then used to find the collector currents

$$I_{Ca} = \frac{\alpha I_E}{1 + e^{-(V_a-V_b)/V_T}}, \tag{42}$$

$$I_{Cb} = \frac{\alpha I_E}{1 + e^{(V_a-V_b)/V_T}}. \tag{43}$$

The base voltages $V_a$ and $V_b$ are still undetermined. However, we will assume the limit $\alpha \to 1$, where the base current goes to zero. In this limit, the two transistor bases may be connected to nodes in an external circuit to set their voltages. As there is no base current, these connections do not affect the voltages in the external circuit. In what follows, we will consider $I_{Ca}$ the output of the differential pair, and label this current $I_o$. Again taking the limit $\alpha \to 1$, we have

$$I_o = \frac{I_E}{1 + e^{(V_a-V_b)/V_T}} = I_E L(-(V_a - V_b)/V_T), \tag{44}$$

where $L(z) = 1/(1 + e^{-z})$ is the standard logistic function. Note that the support circuitry may include a current mirror that inverts the sign of the output current. As this formally has the same effect as a negative value of $I_E$, we will allow $I_E$ to be negative in what follows.

## B.2 Analysis of the logistic regression circuit

As the BJT bases draw negligible current, the voltages $V_{a1}$, $V_{b1}$, $V_{a2}$, and $V_{b2}$ in the circuit can be determined by considering the circuit in the absence of the differential pairs. In this case, we see that (by KCL)

$$R_{a11}^{-1}(V_{C1} - V_{a1}) + R_{a12}^{-1}(V_{C2} - V_{a1}) - R_{a10}^{-1}V_{a1} = 0, \tag{45}$$

and solving for $V_{a1}$ gives

$$V_{a1} = \frac{R_{a11}V_{C1} + R_{a12}V_{C2}}{R_{a11} + R_{a12} + R_{a11}R_{a12}R_{a10}^{-1}} = \frac{R_{a11}^{-1}V_{C1} + R_{a12}^{-1}V_{C2}}{R_{a10}^{-1} + R_{a11}^{-1} + R_{a12}^{-1}}. \tag{46}$$

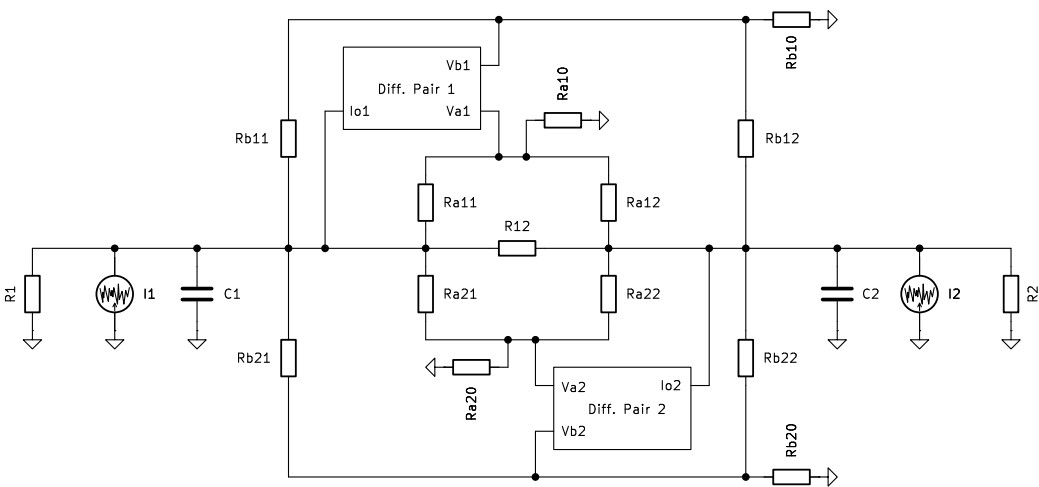

Figure 7: Circuit schematic for the logistic regression posterior sampling device.

The same reasoning applies for $V_{b1}$, resulting in

$$V_{b1} = \frac{R_{b11}^{-1}V_{C1} + R_{b12}^{-1}V_{C2}}{R_{b10}^{-1} + R_{b11}^{-1} + R_{b12}^{-1}}, \tag{47}$$

so

$$V_{a1} - V_{b1} = \frac{g_{a11}V_{C1} + g_{a12}V_{C2}}{g_{a10} + g_{a11} + g_{a12}} - \frac{g_{b11}V_{C1} + g_{b12}V_{C2}}{g_{b10} + g_{b11} + g_{b12}}, \tag{48}$$

where we have written the previous results in terms of the conductance $g = R^{-1}$. The above can be written more conveniently by defining the vectors $\hat{g}_{a1} = (g_{a10} + g_{a11} + g_{a12})^{-1}(g_{a11}, g_{a12})^{\mathsf{T}}$ and $\hat{g}_{b1} = (g_{b10} + g_{b11} + g_{b12})^{-1}(g_{b11}, g_{b12})^{\mathsf{T}}$, in terms of which we have

$$V_{a1} - V_{b1} = (\hat{g}_a - \hat{g}_b)^{\mathsf{T}} V_C. \tag{49}$$

or, defining $\hat{g}_1 = \hat{g}_{a1} - \hat{g}_{b1}$, we simply have

$$V_{a1} - V_{b1} = \hat{g}_1^{\mathsf{T}} V_C. \tag{50}$$

The latter result can be plugged into Eq. (44) to get $I_{o1}$,

$$I_{o1} = I_{E1}L(-\hat{g}_1^{\mathsf{T}} V_C / V_T), \tag{51}$$

where, as before, $L(z) = 1/(1 + e^{-z})$ is the standard logistic function. By an identical derivation to the one above, a similar relation holds for the lower subcircuit

$$I_{o2} = I_{E2}L(-\hat{g}_2^{\mathsf{T}} V_C / V_T). \tag{52}$$

We also assume that all resistors $R_{aij}$, $R_{bij}$ are very large compared to $R_{12}$ so the current flowing through these resistors can be treated as negligible. This assumption does not affect the function of resistors $R_{aij}$, $R_{bij}$ because only the ratios of these resistances determine the voltages $V_{ai}$, $V_{bi}$. Next, we apply KCL to the nodes at the top of capacitors $C_1$ and $C_2$

$$-I_{C1} + I_1 - R_1^{-1}V_{C1} + R_{12}^{-1}(V_{C2} - V_{C1}) - I_{o1} = 0, \tag{53}$$

Similarly, KCL for the node above capacitor $C_2$ reads

$$-I_{C2} + I_2 - R_2^{-1}V_{C2} + R_{12}^{-1}(V_{C1} - V_{C2}) - I_{o2} = 0. \tag{54}$$

Substituting in the expressions derived for the collector currents, we then have

$$-I_{C1} + I_1 - R_1^{-1}V_{C1} + R_{12}^{-1}(V_{C2} - V_{C1}) - I_{E1}L(-\hat{g}_1^{\mathsf{T}} V_C / V_T) = 0, \tag{55}$$

$$-I_{C2} + I_2 - R_2^{-1}V_{C2} + R_{12}^{-1}(V_{C1} - V_{C2}) - I_{E2}L(-\hat{g}_2^{\mathsf{T}} V_C / V_T) = 0. \tag{56}$$

Next we define the conductance matrix

$$\mathcal{G} = \begin{pmatrix} R_1^{-1} + R_{12}^{-1} & -R_{12}^{-1} \\ -R_{12}^{-1} & R_2^{-1} + R_{12}^{-1}, \end{pmatrix}, \tag{57}$$

allowing us to write a single vector equation

$$-I_C + I - \mathcal{G}V_C - I_E L(-\hat{g}^\intercal V_C / V_T) = 0, \tag{58}$$

where we have also set $\hat{g}_1 = \hat{g}_2$. Now using the fact that $dV_C/dt = C^{-1}I_C$, we have the following vector differential equation

$$C\frac{dV_C}{dt} = -\mathcal{G}V_C - I_E L(-\hat{g}^\intercal V_C / V_T) + I. \tag{59}$$

We assume the current vector $I$ has a DC component $I_{DC}$ and a noise component $I_{\text{noise}}$. The noise component is assumed to be an ideal white noise process of infinite bandwidth and power spectral density $S$, which we write $I_{\text{noise}} = \sqrt{S}\xi(t)$. Altogether, we get the stochastic differential equation

$$dV_C = -C^{-1}\mathcal{G}V_C\,dt + C^{-1}I_{DC}\,dt - C^{-1}\alpha I_E L(\hat{g}^\intercal V_C / V_T)\,dt + C^{-1}\sqrt{S}\xi(t)\,dt. \tag{60}$$

Using the identity $\xi(t)\,dt = \mathcal{N}[0, dt]$, this becomes

$$dV_C = -C^{-1}\mathcal{G}V_C\,dt + C^{-1}I_{DC}\,dt - C^{-1}\alpha I_E L(\hat{g}^\intercal V_C / V_T)\,dt + C^{-1}\sqrt{S}\mathcal{N}[0, dt]. \tag{61}$$

At this point it is convenient to define dimensionless quantities which are mapped to the physical parameters of the circuit. Define $\theta = V_C/\tilde{V}$, $\Sigma^{-1} = \tilde{R}\mathcal{G}$, $\Sigma^{-1}\mu = I_{DC}/\tilde{I}$, and $yx = -I_E/\tilde{I}$. Our equation now takes the form

$$\tilde{V}dx = -\tilde{V}\tilde{R}^{-1}C^{-1}\Sigma^{-1}x\,dt + C^{-1}\tilde{I}\Sigma^{-1}\mu\,dt + C^{-1}\tilde{I}L(-\hat{g}^\intercal V_C / V_T)yx\,dt + C^{-1}\sqrt{S}\mathcal{N}[0, \mathbb{I}\,dt]. \tag{62}$$

Next, let $\tau = \tilde{R}C$, and set $\tilde{I} = \tilde{V}C/\tau$ and $S = 2\tilde{V}^2 C^2/\tau$. In this case,

$$d\theta = -\Sigma^{-1}\theta\tau^{-1}dt + \Sigma^{-1}\mu\tau^{-1}dt + L(-\hat{g}^\intercal V_C / V_T)yx\tau^{-1}dt + \mathcal{N}[0, 2\mathbb{I}\tau^{-1}dt]. \tag{63}$$

Finally, we set $\hat{g} = yxV_T/\tilde{V}$ to obtain

$$d\theta = -\Sigma^{-1}(\theta - \mu)\tau^{-1}dt + L(-y\theta^\intercal x)yx\tau^{-1}dt + \mathcal{N}[0, 2\mathbb{I}\tau^{-1}dt], \tag{64}$$

which is identical to Eq. (15)

# C  Analysis of complexity of Gaussian-Gaussian posterior sampling

In this section we analyze the time-complexity of sampling the Bayesian posterior of the Gaussian-Gaussian model using the device in Fig. 1. As shown in Appendix A, the SDE for this circuit is

$$d\theta = -\Sigma^{-1}(\theta - \mu)\tau^{-1}\,dt - \Sigma_{y|\theta}^{-1}(\theta - y)\tau^{-1}\,dt + \mathcal{N}[0, 2\mathbb{I}\tau^{-1}dt], \tag{65}$$

where $\tau = L/\tilde{R}$. This SDE may also be written in terms of the posterior parameters,

$$d\theta = -\Sigma_{\theta|y}^{-1}(\theta - \mu_{\theta|y})\tau^{-1}\,dt + \mathcal{N}[0, 2\mathbb{I}\tau^{-1}dt], \tag{66}$$

where

$$\mu_{\theta|y} = \mu + \Sigma\left(\Sigma + \Sigma_{y|\theta}\right)^{-1}(y - \mu), \tag{67}$$

$$\Sigma_{\theta|y} = \Sigma - \Sigma(\Sigma + \Sigma_{\theta|y})^{-1}\Sigma. \tag{68}$$

The above equation is in the form of a multivariate Ornstein-Uhlenbeck (OU) process [39].

The squared Wasserstein distance between the distribution at time $t$ and the target posterior distribution is [40]

$$W(t)^2 = \|\mu(t) - \mu_{\theta|y}\|_2^2 + D(\Sigma(t), \Sigma_{\theta|y}), \tag{69}$$

where

$$D(\Sigma_1, \Sigma_2) = \text{tr}\left\{\Sigma_1 + \Sigma_2 - 2\left(\Sigma_2^{1/2}\Sigma_1\Sigma_2^{1/2}\right)^{1/2}\right\}. \tag{70}$$

Note that the second term in the squared Wasserstein distance is bounded above as [41]

$$D(\Sigma_1, \Sigma_2) \leq \frac{\|\Sigma_1 - \Sigma_2\|_F^2}{(\sqrt{\lambda_{\min}(\Sigma_1)^2} + \sqrt{\lambda_{\min}(\Sigma_2)})^2} \leq \frac{d\|\Sigma_1 - \Sigma_2\|^2}{(\sqrt{\lambda_{\min}(\Sigma_1)^2} + \sqrt{\lambda_{\min}(\Sigma_2)})^2} \tag{71}$$

Let $\alpha_{\min}$ be the smallest eigenvalue of $\Sigma_{\theta|y}$. At all times $D(\Sigma(t), \Sigma_{\theta|y}$ is bounded above as

$$D(\Sigma(t), \Sigma_{\theta|y}) \leq \alpha_{\min}^{-1} d\|\Sigma_1 - \Sigma_2\|^2 \tag{72}$$

For an OU process the mean behaves as $\mu(t) - \mu_{\theta|y} = e^{-\Sigma_{\theta|y}t/\tau}(\mu(0) - \mu_{\theta|y})$ [39], so the distance between the mean and the target mean is bounded as

$$\|\mu(t) - \mu_{\theta|y}\| = \left\|e^{-\Sigma_{\theta|y}t/\tau}(\mu(0) - \mu_{\theta|y})\right\| \leq e^{-\alpha_{\min}t/\tau}\left\|(\mu(0) - \mu_{\theta|y})\right\|. \tag{73}$$

We now bound the distance between the covariance $\Sigma(t)$ and the target covariance. We use the formula [39]

$$\Sigma(t) = \mathcal{P}_t(\Sigma(0)) + \int_0^t dt' \, \mathcal{P}_{t'}(2\tau^{-1}\mathbb{I}). \tag{74}$$

where $\mathcal{P}_t(X) = e^{-\Sigma_{\theta|y}t/\tau}Xe^{-\Sigma_{\theta|y}t/\tau}$. As $\lim_{t\to\infty}\Sigma(t) = \Sigma_{\theta|y}$, we have

$$\Sigma(t) - \Sigma_{\theta|y} = \mathcal{P}_t(\Sigma(0)) + \int_0^t dt' \, \mathcal{P}_{t'}(2\tau^{-1}\mathbb{I}) - \int_0^\infty dt' \, \mathcal{P}_{t'}(2\tau^{-1}\mathbb{I}) \tag{75}$$

$$= \mathcal{P}_t(\Sigma(0)) - \int_t^\infty dt' \, \mathcal{P}_{t'}(2\tau^{-1}\mathbb{I}). \tag{76}$$

Taking the norm and using the triangle inequality gives

$$\|\Sigma(t) - \Sigma_{\theta|y}\| \leq \|\mathcal{P}_t(\Sigma(0))\| + \int_t^\infty dt' \, \|\mathcal{P}_{t'}(2\tau^{-1}\mathbb{I})\|. \tag{77}$$

As the norm is submultiplicative $\|\mathcal{P}(X)\| \leq e^{-2\alpha_{\min}t/\tau}\|X\|$, so

$$\|\Sigma(t) - \Sigma_{\theta|y}\| \leq e^{-2\alpha_{\min}t/\tau}\|\Sigma(0)\| + 2\tau^{-1}\int_t^\infty dt' \, e^{-2\alpha_{\min}t'/\tau} \tag{78}$$

$$= e^{-2\alpha_{\min}t/\tau}\left(\|\Sigma(0)\| + \alpha_{\min}^{-1}\right). \tag{79}$$

If we would like to have $W(t) \leq W_0$, then we can demand that $\|\mu(t) - \mu_{\theta|y}\|^2 \leq \frac{1}{2}W_0^2$ and $D(\Sigma(t), \Sigma_{\theta|y}) \leq \frac{1}{2}W_0^2$. The inequality involving the mean is satisfied when

$$t \geq \alpha_{\min}^{-1}\tau \ln(\sqrt{2}\|\mu(0) - \mu_{\theta|y}\|W_0^{-1}). \tag{80}$$

The inequality involving the covariance is satisfied when

$$\|\Sigma(t) - \Sigma_{\theta|y}\| \leq \frac{1}{\sqrt{2}}W_0\alpha_{\min}^{1/2}d^{-1/2} \tag{81}$$

This, in turn, is satisfied if

$$e^{-2\alpha_{\min}t/\tau}\left(\|\Sigma(0)\| + \alpha_{\min}^{-1}\right) \cdot \leq \frac{1}{\sqrt{2}}W_0\alpha_{\min}^{1/2}d^{-1/2}. \tag{82}$$

The matrix $\Sigma(0)$ can always be chosen to be zero, so the above inequality becomes

$$e^{-2\alpha_{\min}t/\tau} \leq \frac{1}{\sqrt{2}}W_0\alpha_{\min}^{3/2}d^{-1/2}. \tag{83}$$

We arrive at the requirement

$$t \geq \frac{1}{2}\alpha_{\min}^{-1}\tau \ln\left[\sqrt{2}\alpha_{\min}^{-3/2} \, d^{1/2}W_0^{-1}\right]. \tag{84}$$

Therefore $W(t) \leq W_0$ if the following unified bound is satisfied

$$t \geq \alpha_{\min}^{-1}\tau \max\left(\ln\left[\sqrt{2}\|\mu(0) - \mu_{\theta|y}\|W_0^{-1}\right], \frac{1}{2}\ln\left[\sqrt{2}\alpha_{\min}^{-3/2} \, d^{1/2}W_0^{-1}\right]\right). \tag{85}$$

The quantity $\|\mu(0) - \mu_{\theta|y}\|$ may hide some dependence on dimension, which is discussed presently. It is assumed that the quantity $c = \|\mu_{\theta|y}\|/\sqrt{\alpha_{\max}}$ has an upper bound $c_{\max}$ which is independent of dimension, where $\alpha_{\max}$ is the largest eigenvalue of $\Sigma_{\theta|y}$. That is, the mean of the posterior may be at most $c_{\max}$ standard deviations away from the origin, independent of dimension. This choice represents a particular scaling regime, which we feel is a realistic representation of the accuracy requirements for many applications. We may also choose $\mu(0) = 0$, and this leads to the requirement

$$t \geq \max \alpha_{\min}^{-1} \tau \left( \ln \left[ \sqrt{2} c \sqrt{\alpha_{\max}} W_0^{-1} \right], \frac{1}{2} \ln \left[ \sqrt{2} \alpha_{\min}^{-3/2} d^{1/2} W_0^{-1} \right] \right). \tag{86}$$

In general, the problem may be rescaled in such a way that $\alpha_{\max} \leq 1$, and some rescaling of this kind is realistic given that a particular device will have a specific signal range (that is, the range over which voltages and currents may vary). Redefining the problem this way will also cause the smallest eigenvalue of $\Sigma_{\theta|y}$ to be reduced by a factor of $\alpha_{\max}$, and in this case the bound would be

$$t \geq \max \kappa \tau \left( \ln \left[ \sqrt{2} c \, W_0^{-1} \right], \frac{1}{2} \ln \left[ \sqrt{2} \kappa^{3/2} \, d^{1/2} W_0^{-1} \right] \right), \tag{87}$$

where $\kappa = \alpha_{\max}/\alpha_{\min}$ is the condition number. Subject to these assumptions, we may express the asymptotic time complexity as

$$t = O(\kappa \tau \ln(\kappa^{3/2} d^{1/2} W_0^{-1})) \tag{88}$$

In order to collect $N$ samples the same process is run $N$ times, resulting in complexity

$$t = O(N \kappa \tau \ln(\kappa^{3/2} d^{1/2} W_0^{-1})). \tag{89}$$

# D  Conditioning on multiple I.I.D. samples

When conditioning on a single sample $y$, the energy $U$ can be separated into two terms, one mapping to the prior and the other to the likelihood:

$$U(r) = U_\pi(r) + U_\ell(r), \tag{90}$$

where $\beta U_\pi(r) = -\ln p_\theta(r/\tilde{r})$ and $\beta U_\ell(r) = -\ln p_{y|\theta}(y|r/\tilde{r})$. In general we may have a number of I.I.D. samples $Y = (y_1, \ldots y_N)$, and would like to sample from $p_{\theta|Y}(\theta|Y)$. Because the samples of $y$ are I.I.D., we have

$$p_{Y|\theta}(Y|\theta) = \prod_{i=1}^{N} p_{y|\theta}(y_i|\theta). \tag{91}$$

In this case the likelihood part of the potential energy takes the form

$$\beta U_\ell(r) = -\sum_{i=1}^{N} \ln p_{y|\theta}(y_i|r/\tilde{r}), \tag{92}$$

while the prior part is the same as in the single-sample case, $\beta U_\pi(r) = -\ln p_\theta(r/\tilde{r})$. This form of the potential energy has a convenient physical interpretation: the function $U_\pi$ can be interpreted as the self-energy of the system in state $r$ (that is when it is decoupled from an external system), while the function $U_\ell(\theta)$ can be viewed as an interaction energy between the state $r$ and the state $y$ of an external system. When there are multiple I.I.D. samples, this is analogous to the state $r$ interacting with a collection of external systems in states $Y = (y_1 \ldots y_N)$, and each such interaction contributes its own term to the interaction energy. This provides a framework for building a physical device to sample from the posterior conditioned on multiple I.I.D. samples; one must simply couple a collection of external systems in states $Y = (y_1 \ldots y_N)$ to the system in such a way that each interaction contributes an energy of $-\ln p_{y|\theta}(y|r/\tilde{r})$.

We will now describe another approach to building a physical device that samples from the posterior conditioned on multiple I.I.D. samples of $y$. We first observe that the Langevin equation for the device in this case must be

$$d\theta = \nabla_\theta \ln p_\theta(\theta) \tau^{-1} \, dt + \sum_{i=1}^{N} \nabla_\theta \ln p_{y|\theta}(y_i|\theta) \, dt + \mathcal{N}[0, 2\tau^{-1} dt], \tag{93}$$

As discussed above, if we have a device that can implement the $N$ likelihood drift terms simultaneously then the problem his solved. However, suppose that we have a device that is only capable of implementing a single likelihood term at a time, but $y$ may be varied as a function of time. Additionally, we make the interaction energy for this device larger by a factor of $N$ for reasons that will become clear. That is, we have a device that implements an SDE of the form

$$d\theta = \nabla_\theta \ln p_\theta(\theta)\tau^{-1}\, dt + N\nabla_\theta \ln p_{y|\theta}(y(t)|\theta)\, dt + \mathcal{N}[0, 2\tau^{-1}dt]. \tag{94}$$

We may choose a short time duration $\Delta t$, and set

$$y(t) = y_{\lfloor t/\Delta t\rfloor \bmod N+1}. \tag{95}$$

So for $0 \le t \le \Delta t$ we set $y(t) = y_1$, for $\Delta t < t \le 2\Delta t$ we set $y(t) = y_2$, and so on. Once $t > N\Delta t$ we start over at $y_1$ and continue cycling over all of the I.I.D. samples. Suppose that $\Delta t$ is short enough that all of the samples are cycled over before the state $\theta$ changes significantly. We may then average drift term $N\nabla_\theta \ln p_{y|\theta}$ over a period of time $N\Delta t$ and consider $\theta$ constant within this average. Carrying out this time average, we find

$$\frac{1}{N\Delta t}\sum_{i=1}^{N} \Delta t N \nabla_\theta \ln p_{y|\theta}(y_i|\theta) = \sum_{i=1}^{N}\nabla_\theta p_{y|\theta}(y_i|\theta), \tag{96}$$

resulting in the correct form of the Langevin equation.

# E  Computational complexity of logistic regression

The runtime complexity of digital Langevin sampling of a logistic regression model is $O(n_{\delta t}dN)$, with $n_{\delta t}$ the number of time steps, $K$ the number of trainable parameters, and $N$ the number of data points (in the case of minibatching $b$ replaces $N$. Added to this, there can be some discretization error if the step size is chosen too large, which generally means the number of time steps is made quite large to avoid this (meaning that $n_{\delta t} \gg n_s$). The memory complexity is that of storing the data and the samples, hence is $O(dn_s + N)$. In contrast, running the thermodynamic logistic regression algorithm only includes two digital steps: i) pre-processing and sending over the data to the hardware and ii) initializing the system, which involves setting the prior distribution and the initial state. The gradient evaluations are all done in analog, which incurs a cost of $O(t)$, with $t$ the analog dynamics time. In the best case scenario, where we do not oversample correlated samples, we have $t = n_s\tau_c$, with $\tau_c$ the correlation time. The runtime complexity of the thermodynamic solver is therefore $O(d + N + n_s\tau_c)$, which is a large improvement over the digital case since there is no discretization factor and less multiplicative factors. In addition, note that $t$ can be made extremely small (of the order of the microsecond) in practice thanks to the value of the physical time constants of electronic systems.[1]

---

[1]Since the system is nonlinear, similar bounds to those presented in [27] cannot be obtained.

