# OpenReview forum: "Thermodynamic Bayesian Inference"
_NeurIPS.cc/2024/Workshop/MLNCP — MLNCP Poster_

### Official Review · Reviewer_jdj3 · 2024-10-02

**Rating:** 6
**Confidence:** 3

**Review:**

This paper demonstrated the possibility of using hardware circuit to implement the MCMC sampling algorithm, the Langevin sampling algorithm, by measuring the state of the physical citcuit. In particular, the author proposed the circuit designs from sampling from Gaussian-Gaussian posterior, Bayesian linear regression and Bayesian logistic regression. They demonstrates the effectiveness on low-dimensional problem, and analyze that the time complexity is sub-linear. This is an interesting idea to use hardware the realise the posterior sampling algorithm. However, this raises more questions for its practical usability, which has not been answered in this paper.
1. It seems that there is no principled framework to design the circuit. So if one encountered a complex distribution (e.g. NN), how do you design such the circuit?

2. For comparison, the authors claim that their method is significantly faster than matrix inversion. But a more fair comparison should be the Langevin algorithm simulated in the computer. Since both are Langevin algorithm, can you still gain any computation savings?

3. How does the size of circuit scales with the dataset size and the dimensionality, especially for NN based posterior. Since the neural network typically consist of millions parameters and will be applied to a dataset with large N, it could lead to huge circuits. In addition, since the design of the circuit is dataset specific, the scalability is crucial for practical applicability.

---

### Official Review · Reviewer_DkCt · 2024-10-03
**Good work on  stochastic computing using thermodynamic techniques.**

**Rating:** 7
**Confidence:** 3

**Review:**

Summary:
Overall the authors present an elegantly written manuscript targeting a very important area with an approach that reduces complexity using stochastic computing techniques. The complexity analysis, results, and circuit designs are discussed clearly and make sense. The SPICE simulations have no discussions on energy consumption, current, and voltage ranges, but barring that, the paper is written well.

Strength
- The complex arguments seem to sound up the level of understanding.
- Results are mentioned very clearly and show promise for this approach

Limitations and Questions:
- Is there any particular reason why the Langevin sampling algorithm was selected out of the different sampling algorithms that could be used? Many algorithms allow you to sample from Boltzmann, and if the authors could explain why this specific approach was chosen, it would help the reader. Or is there some intrinsic need for the Langevin dynamics in particular?
- It would be nice to see some more quantitative specifics regarding the circuit. SPICE simulations were mentioned, but nowhere in the paper are ranges of the current source values and noise source peak-to-peak mentioned. It would also be very interesting to understand along with complexity how the energy consumption would look. It feels like a lot of this would be readily available if SPICE simulations were already performed, and would be a great addition to the paper.

---

### Decision · Program_Chairs · 2024-10-10

Accept (Poster)